# Generating a Small Shuttle Vector for Effective Genetic Engineering of *Methanosarcina mazei* Allowed First Insights in Plasmid Replication Mechanism in the Methanoarchaeon

**DOI:** 10.3390/ijms231911910

**Published:** 2022-10-07

**Authors:** Johanna Thomsen, Ruth A. Schmitz

**Affiliations:** Institute for General Microbiology, Christian-Albrechts-University, 24118 Kiel, Germany

**Keywords:** archaeal plasmids, shuttle vector, genetics, rolling circle replication, rep type protein

## Abstract

Due to their role in methane production, methanoarchaea are of high ecological relevance and genetic systems have been ever more established in the last two decades. The system for protein expression in *Methanosarcina* using a comprehensive shuttle vector is established; however, details about its replication mechanism in methanoarchaea remain unknown. Here, we report on a significant optimisation of the rather large shuttle vector pWM321 (8.9 kbp) generated by Metcalf through a decrease in its size by about 35% by means of the deletion of several non-coding regions and the *ssrA* gene. The resulting plasmid (pRS1595) still stably replicates in *M. mazei* and—most likely due to its reduced size—shows a significantly higher transformation efficiency compared to pWM321. In addition, we investigate the essential gene *repA*, coding for a rep type protein. RepA was heterologously expressed in *Escherichia coli,* purified and characterised, demonstrating the significant binding and nicking activity of supercoiled plasmid DNA. Based on our findings we propose that the optimised shuttle vector replicates via a rolling circle mechanism with RepA as the initial replication protein in *Methanosarcina*. On the basis of bioinformatic comparisons, we propose the presence and location of a double-strand and a single-strand origin, which need to be further verified.

## 1. Introduction

*Methanosarcina mazei* belongs to the methanogenic archaea, a monophyletic group of strictly anaerobic archaea that is responsible for the majority of biologically generated methane [1,2]. Since methane is a major greenhouse gas, methanoarchaea have a high ecological relevance [3,4]. Within the methanogenic archaea, *Methanosarcina* species are metabolically versatile and can use diverse substrates such as H_2_+CO_2_, methanol, acetate or methylamines as carbon and energy sources; moreover, they are quite robust towards different environmental stress conditions [5]. This makes *Methanosarcina* especially attractive as a model for studying stress responses, e.g., salt adaptations, the role of small RNAs and small proteins as well as protein complexes [6,7,8]. *M. mazei* is genetically tractable due to the existence of a DNA delivery system, stable replicating plasmids, selection markers, promoters, the ability of forming single colonies on solid agar surfaces and the availability of a fully sequenced genome [9,10]. For gene expression, a shuttle vector was constructed that can propagate in *Escherichia coli* and *Methanosarcina* [11]. For this purpose, plasmid pC2A, which occurs naturally in *M. acetivorans* [12], was fused to an *E. coli* replicon without elucidating the origin and replication mechanism of the generated plasmid in *Methanosarcina*. This led to a large shuttle vector of around 9 kb that is difficult to transfer into *M. mazei* cells due to rather low transformation efficiencies. Moreover, in the case of cloning a large open reading frame or an entire operon into the vector for gene expression, this problem exacerbates accordingly. In order to evaluate the essentiality of regions on the plasmid, detailed knowledge on the replication mechanism is crucial, because plasmids undergo controlled replication independent of the chromosomal DNA [13]. The most common types of plasmid replication mechanisms in bacteria are bidirectional replication (theta-mode or strand-displacement replication) and rolling circle replication [14], which are mostly conducted from plasmids replicating in Gram-negative and Gram-positive bacteria, respectively [15]. Interestingly, only the latter has been postulated for archaeal plasmids [16,17,18]. For pWM321, the rolling circle mechanism has been proposed due to its rep type protein RepA that carries an HUHUU motif similar to other rolling circle-associated proteins [11]. Aiming to reduce the plasmid size, we identified essential regions in the currently used shuttle vector from *Methanosarcina* and successfully generated an optimised shuttle vector by deleting redundant and non-essential regions. One of the essential areas is coding for the replication protein RepA, the biochemical characterisation of which gave further evidence on the proposed replication mechanism by a rolling circle.

## 2. Results

The major goal of this present report was to optimise the currently used shuttle vector pWM321 by decreasing its size and gaining insights into its replication mechanism. To elucidate the necessity of the different plasmid regions, we deleted regions of unknown function within the pC2A-derived replicon and tested the respective stability of the constructs in *M. mazei*. We further cloned and biochemically characterised the gene product of *repA* from pWM321 (in the following referred to as RepA).

### 2.1. Several Shortened Derivatives of the Shuttle Vector Are Stable in M. mazei

#### 2.1.1. Elucidating Crucial Regions for Stable Replication in *M. mazei*

The size of the plasmid pWM321 was decreased step-by-step using a site-directed deletion PCR approach. Therefore, plasmid pRS1452 was constructed from pWM321 by deleting the non-amplifiable puromycin resistance gene (*pac*). Subsequently, a total of six different regions were deleted separately (Table 1 and Figure 1) and the puromycin resistance gene was reintroduced into the various constructs.

After sequence confirmation, all plasmids were transformed using the liposome-mediated transformation procedure into *M. mazei* cells, which were subsequently grown under selective pressure (puromycin presence). Three of the deleted regions turned out to be non-essential for replication as the cultures were growing under selective conditions and the correct plasmid was purified from the cultures and verified by sequencing.

In a second step, all non-essential regions were deleted in parallel, resulting in plasmid pRS1550, which also stably replicated in *M. mazei*. In a last step, unique restriction sites were introduced between the different parts of the plasmid to rearrange the areas according to the modular shuttle vector system from Fink et al. [19]. In addition, a modified *pac* gene with a smaller promoter region was used leading to the even smaller plasmid pRS1595 (Figure 1c), which was also replicable in *M. mazei*.

#### 2.1.2. Smaller Shuttle Vectors Increase the Transformation Success

After successful plasmid construction in *E. coli* and verification of stable replication, we examined the transformation efficiency into *M. mazei* for the respected plasmids using liposome-mediated transformation [11]. According to the protocol, after the transformation no direct plating under selective conditions is possible (Appendix A), which makes the determination of colony forming units impossible. Nevertheless, after applying selective pressure in liquid medium, the cultures transformed with the various shuttle vector constructs showed very different growth behaviours. Cultures transformed with pWM321 usually require 11.8 ± 2.9 days until visible growth appears under selective conditions, while cultures transformed with the smaller constructs pRS1550 and pRS1595 require only 5.8 ± 1.8 and 6.6 ± 0.8 days, respectively (Figure 2a). Moreover, 100% of cultures transformed with either pRS1550 or pRS1595 grow under selective pressure, while only 57 ± 28% of the cultures transformed with pWM321 grow in the presence of puromycin (Figure 2b).

### 2.2. Smaller Shuttle Vectors in M. mazei Show the Same Phenotype and Copy Number as the Original pWM321

To elucidate plasmid stability in *M. mazei*, single colonies were isolated after transformation and the strains were stored in cryo stocks (*M. mazei*/ pWM321, pRS1550 or pRS1595). The strains were also grown in liquid cultures and continuously freshly inoculated every week. After six months (~25 passages), the liquid cultures and the respective strains freshly inoculated from the cryo stock as controls showed no differences regarding the plasmid presence and sequence, indicating that all plasmids stably replicate in *M. mazei* under selective pressure. Furthermore, the growth of the respective strains was monitored for 160 h demonstrating no significant differences in growth behaviour between the parental strain *M. mazei** under non-selective conditions [20] and strains carrying pWM321, pRS1550 or pRS1595 under selective conditions (Figure 3a). The growth rate was determined three times with three biological replicates each, with a nearly identical doubling time between 15.6 and 16.3 h (Figure 3a).

In addition, samples were taken in the lag, early-exponential, late-exponential and stationary phases to compare the overall morphotype of the cells by light microscopy and the respective plasmid copy number by qPCR. While the plasmid copy number of the various plasmids slightly differed during the various growth phases, there was no significant morphological difference of the cells or the copy number of the plasmids pWM321, pRS1550 and pRS1595 detectable in a defined growth phase (Figure 3b,c).

### 2.3. RepA Is the Essential Replication Protein

The *repA* gene was proven to be essential for plasmid stability in *Methanosarcina* (Table 1). The corresponding protein RepA includes a metal binding HUH motif, composed of two histidine residues separated by a bulky hydrophobic residue. This motif is mainly included in the widespread HUH endonuclease superfamily and has key roles in rolling circle replication of plasmids and bacteriophages, as well as plant and animal viruses and is conserved in archaea and bacteria (Figure 4a). As well as the HUH motif, which is mainly involved in metal ion coordination, the basic catalytic mechanism operating in the initiation and termination of rolling circle replication needs an active site Tyr (Y motif) containing either one or two Tyr residues separated by several amino acids [13]. For *M. mazei* RepA, two potential tyrosines (Y301, Y303) have been identified close to the HUH motif (Figure 4b) as well as an FVL of unknown function. Each RepA monomer consists of two domains connected by a short hinge region, whereby the N-terminal origin-binding domain hosts the catalytic activity, and the C-terminal oligomerisation domain is potentially responsible for oligomerisation. The latter consists of four α-helices, whereas the first shows a three-layer α-β-α sandwich fold with antiparallel β-sheets flanked by several α-helices. The active Tyr residue is located in one of these α-helices, while the middle β-strand bears the HUH sequence motif. This overall protein topology is very similar in various bacterial rep type proteins such as the streptococcal RepB from pMV158 [21].

### 2.4. Interaction of RepA and DNA

To investigate its biochemical properties, N-terminal His-tagged RepA was heterologously expressed in *E. coli*, followed by affinity chromatography purification to apparent homogeneity (Figure 4c). Size-exclusion chromatography showed that RepA elutes in its monomeric form with 64 kDa (Figure 4d) in accord with the predicted molecular mass. In agreement with folding transitions measured via Tycho, RepA did not show any significant oligomerisation, leading to the assumption that the purified His-tagged protein does not form oligomers under the tested conditions (His-tag buffer, pH 6.9). It is, nevertheless, likely that the subunits form complexes when incubated with DNA or protein interaction partners. However, no interaction partner was identified using the pull-down method with cell extract from *M. mazei*.

### 2.5. RepA Binds and Nicks Archaeal Plasmids

To further investigate the in vitro interaction of RepA and DNA, purified protein was incubated with different archaeal and bacterial plasmids. The samples were supplemented with a native loading buffer and loaded on an agarose gel. In all cases, RepA bound to the plasmid DNA when incubated at room temperature for 10 min, resulting in a concentration-dependent electrophoretic mobility shift as shown in Figure 5a. As well as the binding of DNA, RepA showed significant nickase activity in the case of the archaeal vectors pWM321 and pRS1595 (Figure 5a, white arrows), but not in the case of the bacterial plasmid (pBluescript (pBSK))—which remained bound but was not nicked—when incubated at 37 °C.

To identify the specific nick site, sequence alignments were performed and showed that the respective shuttle vectors share several sequence motifs with other rolling circle plasmids such as the double-strand origin (DSO) and the single-strand origin (SSO). For the well-characterised bacterial plasmid pMV158, a DSO sequence (5′-GGGGCTACTACGACC-3′) can be found upstream of the rep type gene, while for the archaeal proteins Rep74 and Rep75 the DSO (5′-GGGTTTATCTTG/ATA-3′) is located in the 5′ coding region of the rep type gene, which is a typical feature of RC plasmids [13,16]. *M. mazei repA* has a similar sequence (5′-GGGTTTGCTTCGATG-3′) in the same coding region (Figure 6). SSOs are usually located 100–200 bp upstream of the DSOs. SSO motifs comparable to the ones identified for Rep75 can also be seen for RepA [16]. This putative SSO contains a motif A (5′-AACG-3′), motif B (5′-CnAnnC-3′) and a bacterial primase (DnaG) start site (5′-CTGC-3′) (Figure 6).

To confirm the hypothesis that DSO and SSO of the archaeal shuttle vectors are located in this area, a 345 bp region of pWM321 (ncts. 5088–5432) including the putative DSO and SSO was cloned in pBSK, resulting in pRS1694. This plasmid is nicked by RepA with a slightly higher efficiency than the empty pBluescript vector (after 10 min), but is not as efficient as the shuttle vectors (Figure 5a, right column), indicating that the region is only the partial origin and is missing crucial parts. The area close to the DSO and SSO are often reported to form secondary structures, e.g., hairpins, which might be important for the nick activity and might depend on further regions of the plasmid such as *orf1* or *orf2* encoding potential binding proteins or the region upstream of *repA*. In addition, the nick for pRS1694 is constantly increasing with the protein amount, while too high protein concentrations suppress the nick activity in the case of pWM321 and pRS1595 (Figure 5a).

The nickase activity was also tested in the presence of different metal ions, demonstrating that the nicking activity of pWM321 is optimal in the presence of 10 mM Mg^2+^, while a Ca^2+^ concentration significantly above the physiological level (50 mM) supports a nick in pWM321 and pBSK (Figure 5b). It was shown before that most nucleases require a divalent cation as a cofactor, usually Mg^2+^ or Ca^2+^, and that sequence-specific recognition, DNA binding and phosphodiester cleavage each depends on different metal ions [25,26]. Thus, it seems like RepA is losing its sequence specificity in the presence of Ca^2+^ and causes a nick even in pBSK. The presence of 5–100 mM of the metal ion chelator ethylenediaminetetraacetic acid (EDTA), used as a negative control, resulted in a complete loss of nick activity using any of the tested plasmids. The nick activity was further monitored over a time up to 90 min and this confirmed that nicking activity is significantly higher for the archaeal shuttle vectors pWM321, pRS1595 and pRS1694, including the predicted ori sites, compared to the bacterial plasmid pBSK (Figure 5c). A double exchanged variant of the protein was constructed exchanging two tyrosines in the active centre to phenylalanine (RepA_Y301F_Y303F further referred to as RepA_DM) to identify the active centre of RepA. The activity of RepA_DM was monitored over time and compared to RepA and is slightly less active, although it still shows nicking activity. pBSK is not efficiently cut from either protein variant, while pRS1694 takes a longer time but is—as with the archaeal shuttle vectors—cut almost completely after 90 min (Figure 5c, white arrows). In order to verify the location of the above-mentioned nick site, RepA was incubated with pWM321∆*orf1+2* and pWM321∆*repA*, showing that the nick is slightly less efficient for pWM321∆*repA* (Appendix A). Appropriate controls were used in each case to ensure there was no background activity from other *E. coli* proteins or residual imidazole.

## 3. Discussion

Methanoarchaea are an important key player in resolving the current climate crisis and therefore have been increasingly studied in the last two decades [2,27]. However, great parts of their genetic system remain quite elusive, even though new biotechnological applications for methanoarchaea are on the rise [28]. For instance, the system for protein expression in *Methanosarcina* is long established, but cloning efficiencies are low and details about the replication mechanism of the used shuttle vector remain unknown.

### 3.1. Optimisation of Archaeal Shuttle Vectors for Biotechnological Applications

In the last decades, a few shuttle vectors have been constructed simply by combining an *E. coli* replicon with plasmids that occur naturally in methanogenic archaea such as the plasmid pC2A from *M. acetivorans* [12], or the cryptic plasmids pURB500 from *Methanococcus maripaludis* [29] and pME2001 from *Methanothermobacter marburgensis* [30]. With this approach, several shuttle vectors have been successfully constructed such as pWM321, pDLT44 and pMVS [11,19,31]. Because no naturally occurring plasmids were found in *M. mazei*, pWM321 from *M. acetivorans* was also used for their closely related sister group, leading to problems in *M. mazei* such as insufficient transformation efficiencies or cultures that lose the plasmid after a few generations. We now optimised the archaeal part of pWM321 for the first time to improve genetic engineering and biotechnological applications. Thus, we studied the potential opportunities for shortening the shuttle vector pWM321 in the archaeal part (pC2A), which contains four open reading frames (*repA*, *ssrA*, *orf1* and *orf2*) and two non-coding (nc) regions.

To investigate the essentiality of an area on the vector, we constructed the respective plasmids, lacking one of the above-mentioned regions. The successful transformation of various constructs indicated that nc’, *ssrA*, nc’’’ and parts of nc’’ are not essential for plasmid replication in *M. mazei*. Thus, the final constructs pRS1550 and pRS1595 are up to 35% smaller than the original shuttle vector pWM321 (Figure 1). Cells transformed with the smaller constructs take less time to recover and overall transformation success nearly doubles compared to pWM321 (Figure 2). Based on the observed copy numbers (1–5, depending on the growth phase, Figure 3), the shuttle vectors derived from pWM321 can be considered as low copy plasmids and a strict regulation has to be proposed [32].

The non-essential gene *ssrA* of pCA2 shows some sequence similarity with site-specific recombinases, more specifically, with tyrosine recombinases. This protein family is known to be important for the integration and excision of mobile genetic elements into the host chromosome and the post-replicative segregation of plasmids and circular chromosomes in bacteria, eukaryotes and archaea [33]. Recently, new particular features were highlighted for archaeal tyrosine recombinases as catalysing reactions beyond site-specific recombination such as low-sequence specificity recombination reactions with the same outcome as homologous recombination events [34]. These low-sequence specificity recombinations might be the reason why selected mutants sometimes lose their plasmid but stay puromycin resistant by introducing the *pac* gene into their chromosome. In particular, the deletion of the *ssrA* gene—together with the overall decreased size of the new shuttle vectors—might therefore greatly improve plasmid stability and prevent transformants from losing the plasmid. For low copy plasmids, unlike high copy plasmids, the random segregation of plasmids is often not sufficient to ensure that each daughter cell acquires at least one copy of the plasmid. Thus, most low copy plasmids carry genes that guarantee their maintenance in the population such as a system for efficiently resolving plasmid multimers so that they can be appropriately partitioned [32]. If *ssrA* of pCA2 had such an original function in *M. acetivorans*, it is obviously not essential for plasmid replication (or segregation) in *M. mazei* as the plasmid still stably replicates after the removal of *ssrA* with no changes in plasmid yield, copy number or sequence of the plasmid. Post-replicative plasmid segregation must therefore take place in a sufficient manner, leading to the hypothesis that this task is taken over by one of the chromosomal homologs of SsrA present in different *M. mazei* strains. Interestingly, the intergenic region between *ssrA* and *repA* harbouring two very long direct repeats and a 20 bp perfect inverted repeat immediately following the SsrA coding sequence could be deleted as well. This indicates they are not essential for plasmid replication in *M. mazei*, but may probably play a role in the features that were displayed by SsrA or are part of the non-essential SSO.

The resulting smaller size of the shuttle vector, especially the deletion of *ssrA*, will facilitate and speed up mutant construction in the future. The above-mentioned problems concerning insufficient transformation efficiencies, plasmid stability and cultures losing the plasmid after a few generations were prevented completely when using the smaller constructs, which is an enormous success. Moreover, our findings might hint on how to improve shuttle vectors for other organisms and thus has the potential to promote the use of *Methanosarcina* strains in both research and biotechnological applications. The latter includes the novel possibility to use *M. mazei* as a production strain.

### 3.2. RepA Shows Conserved Motifs of Rolling Circle Initiator Proteins

We have shown that the gene *repA* is essential for plasmid replication and stability in *M. mazei*. Not surprisingly, the corresponding promoter region is also mandatory, which may not only be important for transcription of *repA*, but could also carry important features for plasmid replication such as DSO and SSO or for autoregulation of its own expression. RepA shares homology with the replication initiation proteins, usually referred to as rep type proteins, from a family of phages and plasmids that replicate by a rolling circle mechanism, belonging to the His-bulky hydrophobic residue-His (HUH) endonuclease superfamily [13]. As well as the HUH motif, they usually share a Y motif (Figure 4a), according to which the superfamily is separated into superfamily I and II containing two or one tyrosines in their active site, respectively [35]. Incidentally, some endonucleases from superfamily I also require only one of their Y motif Tyr residues for catalysis, whereas others require both [36]. Particularly interesting is the Y motif of RepA because even though there are two tyrosines (YLY) close to the active site, it can be considered as a part of superfamily II with only one tyrosine (Y301) involved in nicking-closing activities. The same YLY motif can be found in the bacterial pMV158 plasmid family [37].

As with other archaeal rep type proteins, e.g., Rep74 and Rep75, both originating from hyperthermophilic archaea [16,17], RepA has two putative start codons resulting in a size of 46 or 62 kDa. Since only the first CTG codon is preceded by a consensus ribosome-binding site (AGGAA) leading to a 62 kDa protein, this protein was heterologously expressed, purified to apparent homogeneity (Figure 4c) and characterised. As with Rep74 and Rep75, RepA therefore appears to be significantly larger than most of the bacterial rep type proteins described so far (about 40 kDa [17]). As with most replication proteins, RepA has a monomeric form in solution (Figure 4d) and might form multimers upon binding to the origin DNA [13]. Thus, despite the different size, RepA shows high similarity with bacterial rep type proteins such as streptococcal RepB that also consists of an N-terminal origin-binding domain and a C-terminal oligomerisation domain [21].

### 3.3. The Double-Strand and the Single-Strand Origin

The rolling circle replication begins with the sequence-specific cleavage at the double-strand origin conducted by the initiator protein. This cleavage generates a 3′-OH end that allows elongation of the leading strand that is unwound by a host DNA helicase. This is opposite to bidirectional replication (theta-mode and strand-displacement replication), where the replicator recruits initiator proteins in a DNA sequence-specific manner, which results in melting of the DNA helix and loading of the replicative helicase onto each of the single DNA strands [15,38]. Therefore, the ability of heterologously expressed RepA to nick plasmid DNA (Figure 5) strongly suggests that RepA indeed initiates a rolling circle replication process (Appendix A). Here, the tyrosine responsible for the nicking event in vivo remains bound to the 5′-OH of the DNA [13]. It is important to note that the bacterial plasmid pBSK used as a control, carries, in addition to the *ColE1* origin replicating via bidirectional replication, the *f1* origin, which replicates via the rolling circle mechanism [15,32]. pBSK binding of RepA takes place and is as good as in the case of the archaeal plasmids; however, the nicking activity is significantly less efficient, which might be explained by the sequence differences of the *f1* origin, overall indicating that RepA is a rolling circle initiation protein.

Sequence alignments showed that the shuttle vectors pWM321 and pRS1595 share several sequence motifs (DSO and SSO) with other bacterial and archaeal rolling circle plasmids (Figure 6) [13,16]. pBluescript carrying the proposed DSO and SSO (pRS1694) was nicked with a higher efficiency than the empty pBluescript vector and, even though less efficient than the shuttle vectors pWM321 and pRS1595, was cut completely after 60–90 min (Figure 5a). This indicates that the DSO is located in this area, but other regions nearby are likely to also have an effect on the nicking efficiency. Interestingly, in the performed assays, the nicking activity is slightly less efficient when *repA* is absent in the plasmid but is not diminished completely (Appendix A). This indicates that RepA can use alternative DSOs in vitro. However, we currently cannot completely exclude that the identified sequence is, despite the similarities to other archaeal rep type proteins, not the DSO of the shuttle vectors. For several haloarchaeal rolling circle plasmids, for example, a different sequence (5′-TCTCGGCnnnCnnnG-3′) was described as DSO [18] and, interestingly, this sequence matches a sequence (5′-TCTCGGCGAACACCG-3′) in the puromycin resistance gene. As the *pac* gene is present in the *Methanosarcina* shuttle vectors as a selection marker, it might lead to potential alternative nicking sites. After nick of the DSO and the replication, another cleavage leads to the characteristic single-strand DNA (ssDNA) intermediate. This reaction is assumed to be catalysed by the same rep type protein that remained bound to the 5′ end of the parental strand while traveling along with the replication fork [13]. While the first nick can be observed in the assays using RepA, the second nick resulting in ssDNA cannot be detected in vitro as interaction partners such as the helicase, polymerase and other stabilising interaction partners, e.g., the gene products of *orf1* and *orf2* or single-strand DNA binding protein, are missing. In vivo, a single-strand DNA is released (Appendix A), serving as a template for the synthesis of the lagging strand, initiated from the SSO [39]. An SSO motif comparable to the one identified for Rep75 can also be seen in RepA [16] (Figure 6). It is also tempting to speculate that the direct and inverted repeats in the non-coding region of pC2A were the original or alternative SSO. After incorporation of the *E. coli* backbone between those repeats and *repA* during construction of pWM321, they might, however, have lost their function. In the literature, five main types of SSO have been reported so far that are—unlike the DSO—not essential for plasmid replication, provided that an alternative pathway exists for priming lagging strand synthesis [13]. As both nicking and joining are dependent on at least one tyrosine, it is interesting that RepA_DM is still active (Figure 5c), as other proteins were reported to lose their nicking activity after the exchange of the active tyrosine to phenylalanine [18,40,41]. It seems that the identified tyrosines (Y301 and Y303) are not necessarily essential for the nick or their function is carried out by another amino acid after the exchange. Thus, further experiments should shed light on open questions such as the exact molecular mechanism, the catalytic centre and the protein structure of RepA.

## 4. Materials and Methods

### 4.1. Strains and Plasmids

All strains and plasmids used in this study are listed in Table 2. Plasmid DNA was transformed into *M. mazei** by liposome-mediated transformation as described by Ehlers et al. [20]. Plasmid DNA was transformed into *E. coli* DH5α, JM109 λpir or BL21-CodonPlus^®®^-RIL (Stratagene, La Jolla, CA, USA), according to the method described by Inoue et al. [42].

### 4.2. Generation of Plasmids and Construction of Mutant Strains

For the construction of the shortened plasmids, the puromycin resistance gene was first deleted by PCR and an *SbfI* restriction site was introduced by site-directed mutagenesis (SDM). The PCR was followed by *DpnI* digest, T_4_ ligation and heat shock transformation in *E. coli* JM109 λpir. The resulting plasmid pRS1452 served as a template for the following SDM PCRs to delete various regions resulting in seven plasmids each with different deletions on the plasmid (Table 1). Primers were designed to delete the areas ∆526–1287, ∆1316–2287, ∆2325–2778, ∆2811–3845, ∆3908–5688, ∆5712–6149 or ∆29–2556 and ∆5712–6149 (Table 3). Following this, the synthesised puromycin resistance gene was cloned from pRS1513 in the plasmids with *SbfI*, resulting in pRS1532, pRS1515, pRS1531, pRS1518, pRS1522, pRS1523 and pRS1550.

For generation of the modular shuttle vector pRS1595, pRS1550∆*pac* served as template to introduce an *AsiSI* restriction site resulting in pRS1571 by SDM. After successful selection and sequence verification, pRS1571 further served as a template to introduce the restriction sites *FseI* and *SpeI* in an SDM PCR, resulting in pRS1577. Afterwards the *pac* gene and the MCS were synthesised and reintroduced from pRS1585 with *FseI*/*AscI* and from pRS1579 with *AscI*/*SpeI* in two separated cloning steps, resulting in pRS1595. All plasmids were introduced in *M. mazei** as described by Ehlers et al. [20] and transformants were selected based on puromycin resistance. Transformation was verified by plasmid isolation from *M. mazei** and Sanger sequencing of the plasmids. To identify the origin of pWM321, a 345 bp region was amplified using SDM and introduced in pBSK using *ApaI*. For overproduction of RepA, the subsequent gene was amplified from pWM321 via PCR, introduced in pET28a(+) with *NdeI*/*XhoI* and transformed into *E. coli* BL21-CodonPlus^®®^-RIL. For expression of RepA_DM, pRS1559 was used as a template for an SDM PCR and after *DpnI* digest and ligation, resulted in pRS1625.

### 4.3. Growth of M. mazei

*M. mazei* strains were grown under anaerobic conditions without shaking at 37 °C with an N_2_/CO_2_ (80/20) atmosphere in 5 mL or 50 mL minimal medium in closed Hungate tubes or serum bottles, respectively, containing 150 mM methanol as sole energy and carbon source. The medium was supplemented with 100 µg/mL ampicillin to prevent bacterial contamination and with 5 µg/mL puromycin for mutant selection. Growth was generally monitored by determining the turbidity of the cultures at 600 nm and the doubling time was determined in the exponential growth phase. The overall cell morphology was monitored with light microscopy using an Axioskop microscope with an Axiocam 305 color camera (Carl Zeiss AG, Oberkochen, Germany).

### 4.4. Copy Number Determination

Copy numbers of pWM321, pRS1550 and pRS1595 were determined via quantitative PCR. The assays were performed using ViiA™ 7 Real-Time PCR System (Applied Biosystems, Darmstadt, Germany) with the PowerUp™ SYBR™ Green Master Mix (Thermo Fisher Scientific, Waltham, MA, USA) and a primer set that amplifies a 250 bp fragment of *repA*. pWM321, pRS1550 and pRS1595 purified from *E. coli* and concentration determined via Qubit Fluorometer (Thermo Fisher Scientific, Waltham, MA, USA) served as a standard, while for all samples, 1 µL of *M. mazei* culture was used. These cultures were in parallel used for the determination of the cell count using a Thoma cell counting chamber. To determine the copy number per cell, the calculated number of plasmids was divided by the count of cells.

### 4.5. Purification of RepA-His_6_

N-terminal His-tagged RepA was purified from 1 L culture of *E. coli* BL21-CodonPlus^®®^-RIL carrying the plasmid pRS1559 grown in LB at 37 °C under constant shaking. At a turbidity of 0.6 at 600 nm, the protein expression was induced with 100 µM IPTG and the culture was further incubated for 4 h at 37 °C. After harvesting (4.000 g at 4 °C for 20 min) and resuspending in His-tag buffer (50 mM NaH_2_PO_4_, 300 mM NaCl, pH 6.9), the cells were disrupted using a French pressure cell two times (at 4.135 ×10^6^ N m^−2^) followed by centrifugation at 13.000 g at 4 °C for 30 min. The cytosolic supernatant was filtrated through a 0.2 µm filter and incubated in a column with 1 mL Ni-NTA agarose (Qiagen, Hilden, Germany) at 4 °C for 1 h. The column was washed twice with 8 mL His-tag buffer containing 20 mM and 50 mM imidazole before RepA-His_6_ protein was eluted with each 5 × 1 mL His-tag buffer containing 100, 250 and 500 mM imidazole. The fractions were analysed with Coomassie- and silver-staining after separation via a standard 12% SDS-PAGE [44]. The elution fractions containing the purified protein were combined and dialysed using a VISKING^®®^ dialysis tubing (Serva Elektrophoresis, Heidelberg, Germany, MWCO 12-14 kDa) overnight against His-tag buffer. Amicon centrifugal filter (Amicon^®®^ Ultra-4, PLTK Ultracel-PL Membran, 30 kDa, Merck Millipore, Darmstadt, Germany) was used to concentrate the sample. Protein concentration was determined via Bradford assay (Bio-Rad Laboratories Inc., Hercules, CA, USA) according to manufacturer’s protocol.

### 4.6. Size-Exclusion Chromatography

The oligomerisation state of purified RepA-His_6_ was determined by size-exclusion chromatography. The protein was separated in a mobile phase containing His-tag buffer using the ENrich™ SEC 650 column (Bio-Rad Laboratories, Hercules, CA, USA) with a flow rate of 0.5 mL/min. Proteins were detected by monitoring the absorbance at 280 nm and 1 mL fractions were collected. Calibration of the column was performed using the gel filtration standard (Bio-Rad Laboratories, Hercules, CA, USA) containing thyroglobulin (670 kDa), immunoglobulin G (158 kDa), ovalbumin (44 kDa), myoglobulin (17 kDa) and vitamin B12 (1.35 kDa).

### 4.7. Nickase Assay and Electrophoretic Mobility Shift Assay

The ability of RepA-His_6_ to bind and nick plasmid DNA was determined using in vitro approaches. For this purpose, 100 ng of plasmid was mixed with 0–250 ng purified protein in 10 mM Tris buffer (pH 6.9) and 5 mM MgCl_2_ (if not stated otherwise). To identify protein–DNA interaction, the mixture was incubated at room temperature for 10 min, mixed with native loading dye (25% glycerol, 0.05% bromophenol blue) and run on an agarose gel (1%, 30 V, 2 h). To determine the nickase activity of RepA-His_6_, the protein–DNA mixture was incubated at 37 °C. The reaction was stopped by adding stop buffer (30 mM Tris, 300 mM EDTA, 0.1% SDS, 25% glycerol, 0.05% bromophenol blue) and run on an agarose gel (1%, 90 V, 40 min). To investigate the effect of different metal ions, the mixture was supplemented with 5–50 mM MgCl_2_, MnCl_2_, CaCl_2_, ZnSO_4_, NiCl_2_, CuCl_2_ or CoCl_2_. Additionally, the effect of 5–100 mM ethylenediaminetetraacetic acid (EDTA) as chelating agent was examined.

### 4.8. Folding Transition Analysis Using Tycho NT.6

RepA-His_6_ was analysed using Tycho NT.6 (NanoTemper Technologies GmbH, München, Germany) by applying a standard capillary (10 µL) with 0.5 mg/mL protein in His-tag buffer. Thermal unfolding profiles were recorded within a temperature gradient between 35 and 95 °C. In cases where the assays were performed with the addition of plasmid DNA, it was supplemented with 1 µg pWM321 10 min before start and incubated at room temperature or 37 °C.

### 4.9. Pull-Down Analysis

*M. mazei** wild type cells and cells containing plasmid pWM321 were grown at 37 °C without shaking to a turbidity of 0.4 at 600 nm. Cells were harvested and cell-free crude extract was produced as described before [45]. Thirty-fold excess *M. mazei** crude extract was incubated with purified RepA-His_6_ for 10 min at room temperature before mixed with 1 mL His-tag buffer equilibrated Ni-NTA agarose (Qiagen, Hilden, Germany) at 4 °C for 1 h. Non-binding proteins were subsequently washed from the column, whereas RepA-His_6_ and potential interacting proteins were eluted with increasing amounts of imidazole. The fractions were analysed with Coomassie- and silver-staining after separation via a standard 12% SDS-PAGE [44].

### 4.10. Bioinformatic Alignment and Protein Structure Prediction

For protein alignment, the sequence of RepA (AC: AAB39745) from pC2A (AC: NC_002097) of *Methanosarcina acetivorans*, Rep75 (AC: CAA05626) from pGT5 (AC: NC_001773) of *Pyrococcus abyssi*, a hypothetical protein (WP_091938040.1) from *Methanolobus profundi*, a hypothetical protein (WP_048193762.1) from *Methanococcoides methylutens* and the bacterial RepX (P03862) from pC194 (AC: NC_002013) of *Staphylococcus aureus* were used. The alignment was generated with Clustal Omega [46] and amino acids were shaded after Percentage Identity. DSO and SSO sequence alignments were conducted with T Coffee [47] using the above-mentioned sequences and RepB (AC: WP_010889904) of pMV158 (AC: NC_010096) from *Streptococcus agalactiae*. The protein structure of RepA was predicted using AlphaFold2 [48] and visualised using PyMOL (The PyMOL Molecular Graphics System, Version 2.0 Schrödinger, LLC.).

## 5. Conclusions

The optimised shuttle vectors pRS1550 and pRS1595 can stably replicate in *M. mazei* and—because of their reduced size—are easier to transfer into the cell, which will greatly benefit protein expression in *M. mazei* and has the potential to promote the use of *Methanosarcina* strains in research and biotechnological applications. Moreover, our findings on RepA strongly suggest that it is an initiator protein for the rolling circle replication because of the significant in vitro binding and nicking activity of RepA when incubated with the archaeal shuttle vectors pWM321 and pRS1595. We propose the 5′ end of *repA* to be the favoured origin of replication, but alternative origins are conceivable. To our knowledge, the reported results give insights into the replication mechanism of a methanogenic plasmid for the first time and will benefit genetic engineering of methanoarchaea in the future.

## Figures and Tables

**Figure 1 ijms-23-11910-f001:**
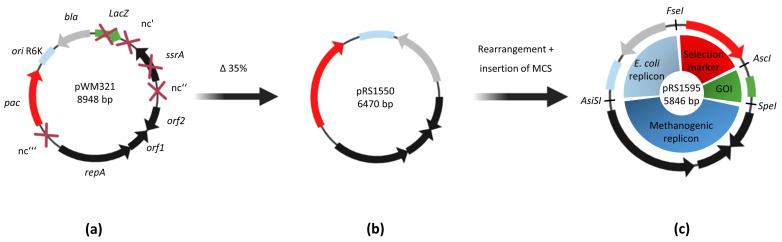
Plasmid maps of pWM321 and the shortest constructs pRS1550 and pRS1595. (**a**) Size reduction overview from pWM321. (**b**) The gene *ssrA* and parts of all non-coding regions from pWM321 were disposable resulting in pRS1550, which was about 2.5 kb smaller size. (**c**) In addition, the restriction enzyme recognition sites *AscI*, *FseI*, *SpeI* and *AsiSI* were introduced and the modules were rearranged resulting in pRS1595. The four modules are the replicon and the selectable marker for *E. coli* (origin *R6K* and *bla* gene), the replicon for *M. mazei* (*repA*, *orf1* and *orf2*), the selectable marker for *M. mazei* (*pac* gene, red) and the gene of interest (GOI, green).

**Figure 2 ijms-23-11910-f002:**
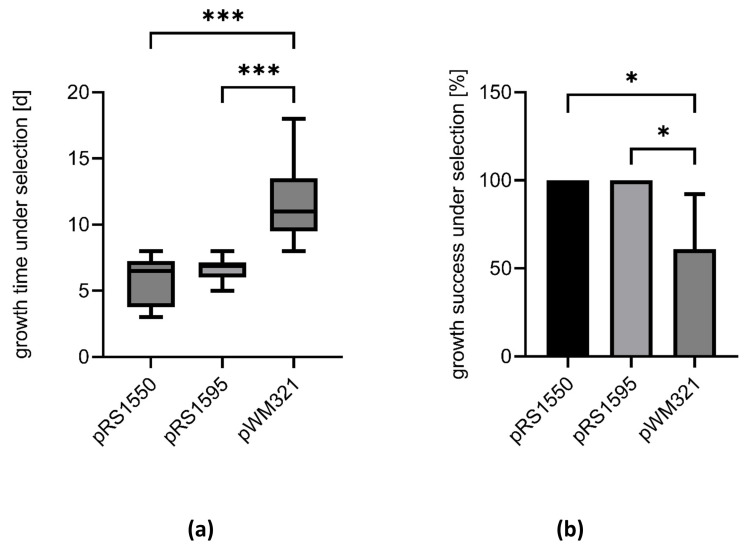
Transformation success in *M. mazei*. The wild type strain *M. mazei** was transformed with pWM321, pRS1550 or pRS1595 using liposome-mediated transformation. (**a**) After the addition of selective pressure, the days were counted until visible growth appeared. (**b**) Growth success of different cultures under selective pressure (puromycin) after transformation. The cultures that have grown have been set in relation to those that have not grown within four weeks. Data represent mean values from 3–6 biological replicates and 2–3 technical replicates each, with standard deviations shown as error bars. The significance was calculated in a one-way ANOVA followed by Dunnett’s multiple comparisons test with p-values resembled as stars with * *p* < 0.05 and *** *p* < 0.001 using GraphPad Prism version 9.4.1 for Windows, GraphPad Software, San Diego, CA, USA, www.graphpad.com (accessed on 11 August 2022).

**Figure 3 ijms-23-11910-f003:**
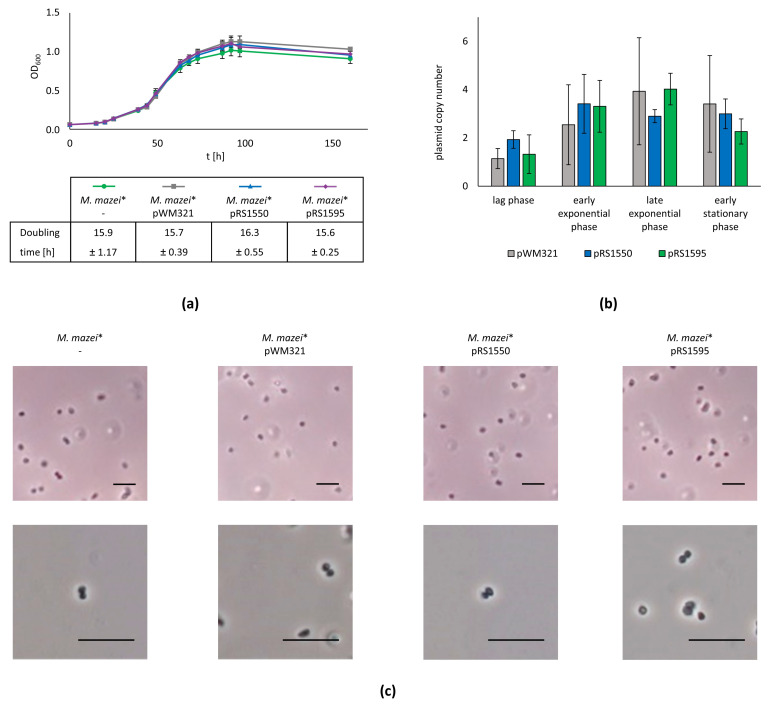
Analysing the effects of various plasmids and their copy number. (**a**) The growth behaviours of a wild type strain and strains carrying pWM321, pRS1550 or pRS1595 were monitored for 160 h. The experiment was carried out three times. Data represent mean values from one representative run containing three biological replicates each, with standard deviations shown as error bars. (**b**) The plasmid copy number was determined via qPCR during lag, early-exponential, late-exponential and stationary phase (24, 48, 72 and 96 h after inoculation, respectively). The experiment was carried out three times. Data represent mean values from one representative run containing three biological replicates, each measured in duplicate, with standard deviations shown as error bars. (**c**) The overall cell morphology was monitored using light microscopy. The scale bar represents 10 μm.

**Figure 4 ijms-23-11910-f004:**
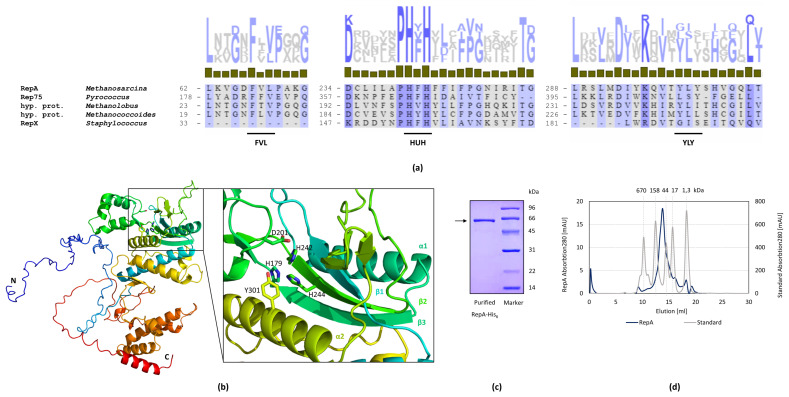
Characterisation of RepA. (**a**) Alignment of the FVL, HUH and YLY motif of putative rep type proteins (indicated with black lines). The amino acid alignment of the archaeal proteins RepA from *Methanosarcina acetivorans*, Rep75 from *Pyrococcus abyssi* [16], putative rep type proteins from *Methanolobus profundi* and *Methanococcoides methylutens* [22] and the bacterial RepX from *Staphylococcus aureus* [23] was generated with Clustal Omega and amino acids were shaded after Percentage Identity (grey indicating low PID, blue indicating high PID). The proteins show high similarity in the region of the HUH motif. (**b**) Predicted secondary structure of RepA. The overall structure of RepA is shown as a cartoon model. Each RepA monomer consists of the N-terminal origin-binding domain, carrying an α-β-α sandwich fold with the active centre and the C-terminal domain that is potentially important for oligomerisation. The amino acids proposed to be part of the active centre are shown as sticks including the HUH motif (H242, F243, H244) and YLY motif (Y301). The structure was predicted with AlphaFold2 and visualised using PyMOL. (**c**) Coomassie-stained SDS-PAGE of heterologously expressed and purified His-tagged RepA. RepA was overexpressed in *E. coli* BL21 (DE3)-Codonplus-RIL containing pRS1559 by induction with 100 μM IPTG in LB medium for 4 h and purified by Ni-NTA chromatography. The LMW marker from the Amersham LMW Calibration Kit for SDS Electrophoresis was used as ladder. (**d**) Size-exclusion chromatography (SEC) of RepA. SEC revealed that RepA elutes in its monomeric form (64 kDa) in His-tag buffer. The SEC Standard (Bio-Rad, #1511901, MW 1350–670,000) was used as a standard.

**Figure 5 ijms-23-11910-f005:**
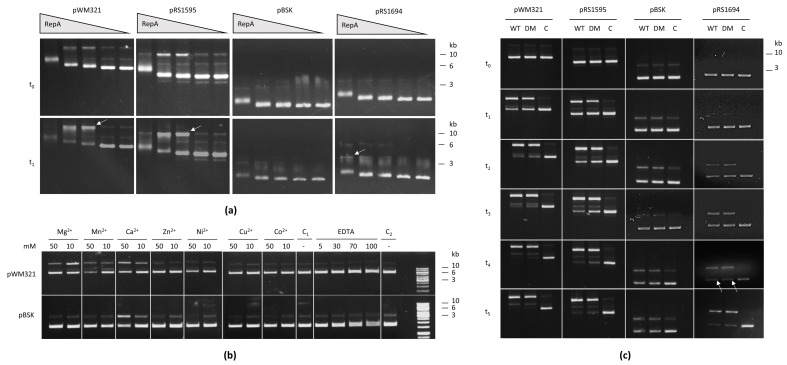
Binding and nicking activity of RepA of archaeal shuttle vectors and bacterial plasmids. (**a**) An electrophoretic mobility shift assay was performed on 1% agarose. In each case, 100 ng plasmid and 250, 100, 50, 10 and 0 ng purified RepA (from left to right) was used in 10 mM Tris (pH 6.9, 500 µM MgCl_2_). The archaeal shuttle vectors pWM321 and pRS1595 were compared with the bacterial plasmid pBSK and pRS1694, pBSK containing a 345 bp fragment of the archaeal replicon. The mixtures were incubated at room temperature for 10 min and subsequently loaded on a gel (t0) in native buffer (25% glycerol, 0.05% bromophenol blue). The remaining assays were further incubated for 10 min at 37 °C before samples were taken (t1). The gel was run for 2 h at 30 V. (**b**) The effect of different metal ions on the activity of RepA was investigated. In each case, 100 ng of plasmid and 20 ng of protein were used in 10 mM Tris (pH 6.9). The mixtures were supplemented with 10 or 50 mM of different divalent metal ions or 5–100 mM EDTA. Control 1 was a mixture of plasmid and RepA without supplementation, while control 2 was only plasmid without RepA. After incubation at 37 °C for 30 min, stop buffer (30 mM Tris, 300 mM EDTA, 0.1% SDS, 25% glycerol, 0.05% bromophenol blue) was added and samples were loaded on a 1% agarose gel. (**c**) The nicking activity of RepA (WT) and RepA_Y301F_Y303F (DM) was compared for pWM321, pRS1595, pBSK and pRS1694. In each case, 100 ng of plasmid and 20 ng of protein was used in 10 mM Tris (pH 6.9, 10 mM MgCl_2_). The mixtures were incubated at 37 °C and samples were taken after 0, 15, 30, 45, 60 and 90 min. For each condition, a control C without RepA was performed.

**Figure 6 ijms-23-11910-f006:**
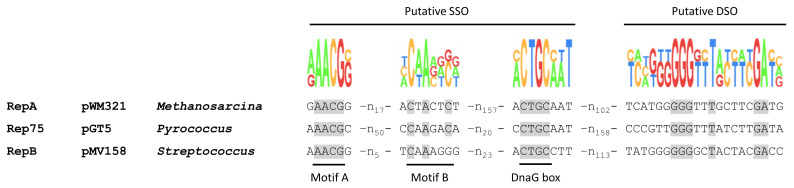
Identification of putative pWM321 and pRS1595 double-strand origin and single-strand origin. Alignment of the sequence of the putative DSO and SSO of RepA from pWM321 and pRS1595 with the according sequence of Rep75 from the archaeal plasmid pGT5 from *Pyrococcus abyssi* [16] and of RepB from the bacterial plasmid pMV158 from *Streptococcus agalactiae* [24]. Putative motifs A, B and the DnaG box as primase binding site of bacterial phages and plasmids are indicated in the SSO. The DSO consists of just one motif, 100–200 nucleotides downstream of the SSO. Conserved nucleotides are shaded grey.

**Table 1 ijms-23-11910-t001:** Plasmid derivatives. Each plasmid misses one (or in the case of pRS1550, several) region(s) of the original shuttle vector pWM321. After construction in *E. coli* and verification, the plasmids were transformed in *M. mazei*, with ‘+’ representing a successful transformation and stable replication and ‘–‘ meaning no transformants were selectable.

Name	Size (kbp)	Deleted Region	Area in pWM321	Stability in *M. mazei*
pRS1532	~8.2	∆nc‘^1^	∆526–1287	+
pRS1515	~8.0	∆*ssrA*	∆1316–2287	+
pRS1531	~8.4	∆nc‘‘^1^	∆2325–2778	–
pRS1518	~7.9	∆*orf1+2*	∆2811–3845	–
pRS1522	~7.2	∆*repA*	∆3908–5688	–
pRS1523	~8.5	∆nc‘‘‘^1^	∆5712–6149	+
pRS1550	~6.5	∆nc‘ + ∆*ssrA* + parts of ∆nc‘‘^1^ ∆nc‘‘‘^1^	∆29–2556 ∆5712–6149	+
pRS1595	~5.8	∆nc‘ + ∆*ssrA* + parts of ∆nc‘‘ ∆nc‘‘‘^1^	∆29–2556 ∆5712–6149	+

^1^ nc = non-coding region.

**Table 2 ijms-23-11910-t002:** Strains and plasmids used in this study.

Strain/Plasmid	Genotype/Relevant Characteristics	Source/Reference
*M. mazei**	Potential cell wall mutant	[20]
*E. coli* DH5α	General cloning strain	[43]
*E. coli* JM109 λpir	General cloning strain	[43]
*E. coli* BL21-CodonPlus^®®^-RIL	Overexpression strain with broader codon usage	Stratagene, La Jolla, CA, USA
pWM321	Shuttle vector *E. coli*–*Methanosarcina* (8.9 kbp)	[11]
pET28a(+)	expression vector, N-term. His-tag	Novagene^®®^, Merck Millipore, Darmstadt, Germany
pRS1452	pWM321∆*pac*	This work
pEX-K248_pac_JT	Synthesised *pac* gene under the control of *pmcrB*	Eurofins Genomics Life Science Services, Ebersberg, Germany
pRS1515	pRS1452∆*ssrA*+*pac*	This work
pRS1518	pRS1452∆*orf1+2*+*pac*	This work
pRS1522	pRS1452∆*repA*+*pac*	This work
pRS1523	pRS1452∆nc‘‘‘+*pac* ^1^	This work
pRS1531	pRS1452∆nc‘‘+*pac* ^1^	This work
pRS1532	pRS1452∆nc‘+*pac* ^1^	This work
pRS1550	pRS1452∆29–2556, ∆5712–6149+*pac*	This work
pRS1559	pET28a(+)*repA*	This work
pEX-K168_MCS_JT	Synthesised MCS	Eurofins Genomics Life Science Services, Ebersberg, Germany
pEX-K248_pac2_JT	Synthesised *pac* gene under the control of *pmcrB*	Eurofins Genomics Life Science Services, Ebersberg, Germany
pRS1595	Modular shuttle vector	This work
pRS1625	pET28a(+)*repA_DM*	This work
pBluescript II KS (+)	General cloning vector	Stratagene, La Jolla, CA, USA
pRS1694	pBSK with 345 bp region of pWM321 (ncts. 5088–5432)	This work

^1^ nc = non-coding region.

**Table 3 ijms-23-11910-t003:** Primers used in this study.

Plasmid	Forward Primer	Reverse Primer	Purpose
pRS1452	5′-CGCCCGCCCCACGAC-3′	5′-CCTGCAGGTTTTGATGTAGTTTCTTACTAC-3′	∆6190–7150 from pWM321
pRS1532	5′-GATCCCGCAGATTATGGAAC-3′	5′-CAATTTCACACAGGAAACAGC-3′	∆526–1287 from pWM321
pRS1515	5′-GTATGTAAATAAATACTTTGTGC-3′	5′-GAAATAATGTTCCATAATCTGC-3′	∆1316–2287 from pWM321
pRS1531	5′-TTGTCGAAGAACTTCCAAAC-3′	5′-TAAATGACATCTATGCACAAAG-3′	∆2325–2778 from pWM321
pRS1518	5′-CGTATCACTTTAGGCTTTAAG-3′	5′-GATCGGTCTACTGTTTGGAAG-3′	∆2811–3845 from pWM321
pRS1522	5′-GAATAAGATTAACGCCTACC-3′	5′-CGTTCAACAAGGCTTTTG-3′	∆3908–5688 from pWM321
pRS1523	5′-CACTATCAAATGACATTGTAGTAAG-3′	5′-TAAGGTAGGCGTTAATCTTATTC-3′	∆5712–6149 from pWM321
pRS1550	5′-GTCACAACATTCACAAAAATAG-3′ 5′-CACTATCAAATGACATTGTAGTAAG-3′	5′-CCTGAATGGCGAATGGTTAAGG-3′ 5′-TAAGGTAGGCGTTAATCTTATTC-3′	∆29–2556 and ∆5712–6149 from pWM321
pRS1571	5′-GCGATCGCAACCTGCAGGTTCACTG-3′	5′-GATGTAGTTTCTTACTACAATGTC-3′	Introduce *AsiSI* site in pRS1550
pRS1577	5′-GGCGCGCCTTAACTAGTCGCCATTCAG-3′	5′-GGTGGCACTGGCCGGCCAAATGTGCGCG-3′	Introduce *FseI*/ *SpeI* site in pRS1550
pRS1559	5′-CGACAGGAAATGTCATATGAGTTCTGATTTTAG-3′	5′-GCCCAAAAGCCTTCTCGAGCGTGGCATCTC-3′	RepA overexpression
pRS1625	5′-GAGAGAAAAGATAAGTCACCTGC-3′	5′-CTTTTCTCTCACGTTGGGCAG-3′	RepA _DM overexpression
pRS1694	5′-GTTTTGGGCCCGGTTCGC-3′	5′-CTCCCGGGCCCAAGTCCATCGAAGC-3′	Introduce putative DSO/ SSO in pBSK

## Data Availability

Not applicable.

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
