# Peer review of "Generating a Small Shuttle Vector for Effective Genetic Engineering of Methanosarcina mazei Allowed First Insights in Plasmid Replication Mechanism in the Methanoarchaeon"

_ijms, 2022, doi:10.3390/ijms231911910_

Round 1
Reviewer 1 Report
The manuscript by Johanna Thomsen and Ruth A. Schmitz reports an optimization of a shuttle vector derived from methanogenic archaean. By deleting unessential regions in the plasmid DNA, size of the plasmid is reduced by 35% and the transformation efficiency is greatly enhanced, while the copy number and growth of the strain harboring the shortened plasmid were not affected. The study also characterized the essential RepA protein, and the results support that the plasmid is replicated in a rolling cycle fashion. Although the methodology adopted in this study is classical, the study provides a timely genetic tool for the the genetic analysis and biotechnological applications of methanogens. The paper is well written with only a concern about the necessity of Figure 7 as a figure in the main text. I would suggest it to be placed as a supplementary figure. Otherwise, I think this a nice article deserved to be published in The International Journal of Molecular Sciences.
Author Response
Response to Reviewer 1 Comments
Point 1: The paper is well written with only a concern about the necessity of Figure 7 as a figure in the main text. I would suggest it to be placed as a supplementary figure.
Response 1: We thank Reviewer 1 for the supportive judgement on our manuscript and for evaluating it as ‘well written'.
We agree with Reviewer 1 that Figure 7 is not essential for the main text and therefore placed it as a supplementary figure.
Reviewer 2 Report
The manuscript presented here is comprehensive and carefully conducted. The manuscript is well written and all data presented are backed on the data generated. This reviewer has no further remarks to improve the quality of the manuscript.
Author Response
Response to Reviewer 2 Comments
Point 1: The manuscript is well written and all data presented are backed on the data generated. This reviewer has no further remarks to improve the quality of the manuscript.
Response 1: We thank Reviewer 2 for the supportive judgement on our manuscript and for evaluating it as ‘well written'. According to the concern of Reviewer 1 we placed Figure 7 as a supplementary figure.